# Efficiency of a music-based intervention as an adjunct to the first noninvasive ventilation session in acute exacerbation of COPD: A randomized single-blind controlled trial protocol

Imen Ben Saida[1‡]*, Dhekra Chebil[2‡], Wassim Jomaa[3], Marwa Zghidi[1], Khalil Attia[1], Mohamed Boussarsar[1]

**1** Faculty of Medicine of Sousse, Farhat Hached University Hospital, Medical Intensive Care Unit, Research Laboratory LR12SP09 "Heart Failure", University of Sousse, Sousse, Tunisia, **2** Preventive Medicine Department, Ibn Al Jazzar University Hospital, Research Laboratory LR19SP01, "Measurement and Support for the Performance of Healthcare Establishments", Faculty of Medicine of Sousse, University of Sousse, Kairouan, Tunisia, **3** Higher Institute of Musicology of Sousse, University of Sousse, Sousse, Tunisia

☯ These authors contributed equally to this work.
‡ These authors share first authorship on this work.
* imen.bensaida@yahoo.com

## Abstract

### Introduction

Noninvasive ventilation (NIV) is a cornerstone in the management of Acute Exacerbation of Chronic Obstructive Pulmonary Disease (AECOPD). The aim of this study is to assess the impact of music-based intervention on the efficiency of NIV in patients with AECOPD.

### Methods

It's a prospective, randomized, single-blinded, parallel-group trial. Critically ill COPD patients admitted for acute exacerbation and eligible for NIV will be included. Patients in the control group will receive only NIV. In the intervention group, patients will receive NIV with music-based intervention. The music will be delivered through headphones. Both groups will receive standard care in accordance with current clinical guidelines. The primary endpoint is the change in arterial Pressure of Carbon dioxide PaCO2, defined as the difference between baseline $PaCO_2$ (prior to NIV initiation) and $PaCO_2$ measured after 4 hours. Secondary endpoints will be recorded at different time points: change of PaCO2 over time ($h_0$, $h_2$, $h_4$), unplanned interventions, changes in respiratory and cardiovascular parameters ($h_0$, $h_2$, $h_4$), changes in Respiratory comfort ($h_0$, $h_2$, $h_4$), Borg Dyspnea Scale ($h_0$, $h_2$, $h_4$), Verbal Numeric Rating Scale for pain assessment ($h_0$, $h_2$, $h_4$), Encephalopathy score ($h_0$, $h_2$, $h_4$), Richmond

**Data availability statement:** No datasets were generated or analysed during the current study. All relevant data from this study will be made available upon study completion.

**Funding:** The author(s) received no specific funding for this work.

**Competing interests:** The authors have declared that no competing interests exist.

Agitation-Sedation Scale ($h_0$, $h_2$, $h_4$), and psychological assessment (Rapid Clinical Test For delirium, State Trait Anxiety Inventory, perceived stress scale) ($h_0$, $h_4$).

## Discussion

This study is expected to contribute reliable scientific evidence for the optimal management of AECOPD, potentially reducing the need for intubation and mechanical ventilation and their associated complications.

## Trial registration

The study was prospectively registered on the Pan African Clinical Trial Registry within the number PACTR202501862862010.

## Introduction

Chronic obstructive pulmonary disease (COPD) is a major global health concern, ranking as the third leading cause of death worldwide [1]. It involves a complex interplay of inflammation and structural changes in the airways and lung parenchyma, leading to airflow limitation, impaired gas exchange, and acute exacerbations [2]. Non-invasive ventilation (NIV) is a cornerstone therapeutic strategy for managing patients with severe exacerbations, particularly those with hypercapnic encephalopathy [3]. Efficiency of NIV in acute exacerbation of COPD (AECOPD) patients with hypercapnic encephalopathy refers to how quickly and effectively NIV achieves its intended physiological effect of improving gas exchange. It is worth noting that an improvement in gas exchange is a strong predictor of a successful outcome with NIV; in patients who respond, this improvement is almost universally observed within the first 2–4 hours after NIV initiation [4,5]. However, the efficiency of this technique is dependent on several factors. In fact, patient anxiety and discomfort, despite appropriate pressure settings and equipment, can lead to premature mask removal and excessive air leaks [6]. Such complications not only diminish the intervention's efficiency but also complicate clinical management, necessitating additional support from healthcare personnel to reassure and stabilize the patient [7]. The clinical challenges of NIV intolerance sometimes lead to light sedation for comfort and interface management [8]. While potentially helpful, sedation carries risks like respiratory depression, hypotension, and delirium [8]. Therefore, non-pharmacological interventions may be of great interest [9].

In recent years, there has been growing interest in applying music-based intervention in medical contexts, particularly in intensive care units. Recent studies have explored the potential benefits of music-based intervention in patients with acute respiratory failure requiring invasive mechanical ventilation [10–12]. Some studies have demonstrated positive effects of music-based intervention on anxiety, pain, analgesic use, delirium, and overall patient satisfaction [10,11,13–15]. Music-based intervention can also reduce agitation in confused patients, improve mood, and facilitate communication [10]. Within the context of NIV, music-based intervention may

serve as a valuable non-pharmacological intervention. By enhancing patient acceptance and tolerance of the technique, music-based intervention can potentially improve the efficiency of NIV sessions [16]. Soft, rhythmic melodies can alleviate anxiety and agitation associated with hypercapnic encephalopathy, promoting better patient cooperation. Music can also distract patients from unpleasant NIV sensations, and for disoriented patients, it can facilitate non-verbal communication. Additionally, some studies suggest music can influence breathing rate, potentially aiding adaptation to NIV [17,18].

There is limited literature on the impact of music-based intervention on NIV experience. A qualitative case series by Davies et al. [18] suggested that music-based intervention can facilitate the transition to NIV in patients with motor neuron disease by providing distraction from the machine and promoting relaxation. However, only one study has evaluated the effect of music-based intervention on critically ill patients requiring NIV. Messika et al. [16] found no significant effect of a 30-minute musical intervention on the tolerance and efficiency of NIV in acute respiratory failure. To the best of our knowledge, no studies have yet examined the intra-session effects of music-based intervention on critically ill COPD patients undergoing NIV for hypercapnic encephalopathy.

In managing severe AECOPD with NIV, the primary aim is to offset the increased resistive load imposed by the exacerbation. Physiologically, the total pressure required for breathing (Ptot) is the sum of the pressure generated by respiratory muscles (Pmusc) and the pressure applied to the airways (Paw). The goal of NIV is to balance the recruitment of maximal Pmusc without inducing fatigue, achieved by providing adequate pressure support (PS) while minimizing leaks. We hypothesize that this equilibrium and the overall efficiency of NIV can be enhanced by incorporating music. Music, particularly with a tempo between 60 and 80 bpm, has been shown to significantly affect respiration by promoting relaxation and stabilizing vital signs [19,20]. It stimulates the limbic system, leading to endorphin release and a sense of well-being [20,21]. Furthermore, some research suggests music can influence serotonin production, which may enhance respiratory patterns, leading to improved breathing rates and depth [21]. A systematic review and metanalysis by Huang et al. [22] suggested that music-based intervention can enhance potential therapeutic effects in COPD patients.

By addressing the psychological and emotional barriers to effective NIV (anxiety, discomfort, poor tolerance), intra-session music-based intervention could potentially improve patient acceptance and adherence to the technique. This, alongside the direct physiological effects of music on the respiratory system and vital signs, can in turn contribute to a more relaxed and efficient respiratory pattern, indirectly enhancing ventilation efficiency and arterial partial pressure of carbon dioxide ($PaCO_2$) clearance.

## Objective

Based on this conceptual framework, we aim to assess whether incorporating an intra-session music-based intervention during the first NIV session for severe AECOPD management enhances its efficiency in reducing $PaCO_2$ as a primary outcome, compared to standard NIV therapy.

## Hypothesis

We hypothesize that incorporating a 2-hour music-based intervention during the initial NIV session in adults hospitalized with AECOPD will result in a significantly greater reduction in $PaCO_2$ after 4 hours of NIV, compared to standard NIV care alone, supporting the superiority of the intervention over standard care.

## Materials and methods

### Design

This is a mono-center, prospective, randomized, single-blinded, parallel-group trial involving critically ill patients admitted to the ICU for AECOPD and requiring NIV.

The expected start and end dates for this study are June 2025 and January 2026, respectively.

This protocol has been developed in accordance with the Standardized Protocol Items: Recommendations for Interventional Trials (SPIRIT) guidelines and checklist (S1 File) (https://spirit-statement.org/).

The study was prospectively registered on the Pan African Clinical Trial Registry within the number PACTR202501862862010.

## Study setting

Recruitment will take place in a 12-bed medical Intensive Care Unit (ICU) at Farhat Hached Hospital in Sousse, Tunisia. The medical staff comprises a department chief, three associate professors, three assistant professors, 12 ICU residents, and 4 interns supported by a paramedical team of 26 nurses and 2 physiotherapists. This unit primarily admits patients with AECOPD. A 4-bed unit is dedicated to NIV, with individualized rooms operating since the year 2000. NIV is a standard treatment modality within this unit and is routinely used for eligible patients according to established clinical guidelines.

## Study population

Patients admitted with AECOPD will be assessed for eligibility. Screening and enrolment will proceed until the target number of participants is reached.

### Eligibility criteria.

- *Inclusion criteria:* Consecutive adult patients (≥18 years) with COPD admitted to the ICU for an AECOPD will be included. Eligibility requires meeting the GOLD criteria for COPD, having a pH < 7.35, and being assessed as suitable for NIV by the attending physician.

- *Exclusion criteria:*

  - Patients unable to tolerate NIV or for whom NIV is contraindicated

  - Patients with severe hearing impairment or deafness

  - Patients for whom life-sustaining therapies have been withdrawn

  - Patients who have received antidepressant or antipsychotic medications in the previous two months

  - Aversion of music as assessed by personal interview

  - Pregnancy

  - Enrollment in other trials

**Informed consent.** Written informed consent will be obtained from all eligible participants. The researcher will explain the study to each potential participant, provide them with an informed consent form, and answer any questions they may have. Participants will then be given 10 minutes to consider their participation. Written informed consent will be obtained from each participant prior to enrolment. For participants who are unable to provide consent, a legally authorized representative (e.g., a family member) will provide written informed consent on their behalf.

## Randomization

Prior to enrolment, all inclusion and exclusion criteria will be thoroughly assessed. Eligible participants will be randomly assigned in a 1:1 ratio to either the intervention group (music-based intervention) or the control group (headphones without music). This randomization will be conducted using a web-generated blocked randomization sequence via the

Research Randomizer platform ([www.randomizer.org](http://www.randomizer.org)), designed to maintain balanced group sizes throughout the trial. The randomization sequence will be generated in advance by a research assistant independent of participant recruitment, clinical care, outcome assessment, or data analysis.

**Concealment mechanism.** To ensure strict allocation concealment and prevent allocation bias, the randomization sequence will be securely stored and inaccessible to all study investigators and site staff involved in screening, enrolment, or intervention delivery. Group assignments will be concealed using a series of sequentially numbered, opaque, sealed envelopes prepared in advance. Each envelope will contain the participant's assigned group and a unique identifier code. Following the acquisition of written informed consent and confirmation of eligibility, a designated research nurse, who is not involved in participant recruitment or care, will open the corresponding sequentially numbered envelope to reveal the group allocation. This individual will be the sole person authorized to handle the sealed allocation envelopes. This procedure ensures allocation concealment until the point of randomization, maintaining methodological integrity and minimizing allocation bias.

The flow chart of the investigation is illustrated in Fig 1.

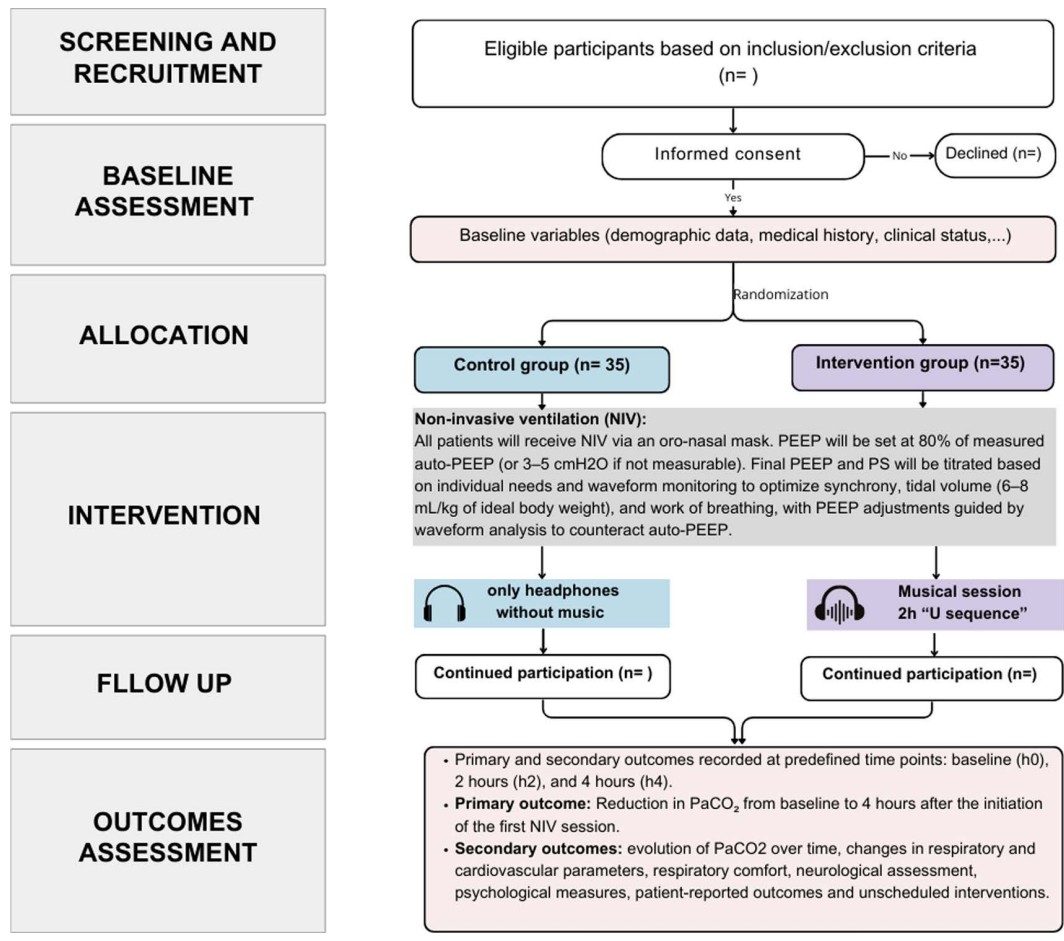

**Fig 1. Flow chart of study process.**

## Blinding

Given the nature of the intervention, blinding of participants is not feasible, as they will know whether they are listening to music. Nevertheless, the study will follow the principles of the Prospective Randomized Open Blinded Endpoint (PROBE) design [23], which is specifically intended to minimize bias in outcome assessment and data analysis.

To maintain blinding integrity and minimize bias, only the participant and the designated research assistant responsible for opening the allocation envelope and administering the assigned intervention will be aware of the group assignment. This individual will not be involved in outcome assessment, clinical follow-up, or data analysis. All outcome assessors, trained residents not involved in randomization or intervention delivery, will remain blinded to group allocation throughout the study. Similarly, data analysts will be blinded to treatment assignment to ensure objectivity in statistical analyses and interpretation of results.

Participants will be explicitly instructed not to disclose their group allocation to any member of the clinical or research team involved in follow-up or outcome assessment. This approach is designed to preserve the integrity of the blinding process and uphold the internal validity of the trial.

## Study intervention

All patients will be equipped with headphones and will receive either a 2-hours session of music-based intervention or will keep the headphones on without music. Both groups will receive NIV as part of standard care.

**Standard of care.** All patients will receive NIV delivered through an oro-nasal mask (Fisher & Paykel) with a seal adapted to the patient's face, in a semi-recumbent position with the door closed and a calm environment, using either a Mindray SV 600 or a Lowenstein Elisa 600 ventilator, without an artificial nose or humidifier to limit instrumental dead space. PEEP will be set at 80% of auto-PEEP measured before the start of NIV (using a short expiratory pause of <0.3 seconds); if auto-PEEP cannot be measured, initial PEEP will be 3–5 cmH2O. Final PEEP and PS will be titrated based on individual needs and continuous ventilator waveform monitoring to optimize synchrony, tidal volume (target 6–8 mL/kg ideal body weight), and work of breathing. PEEP will be adjusted to counteract auto-PEEP, guided by waveform analysis [4,24]. Both groups will receive standard AECOPD therapy, which includes oxygen titrated to maintain a target oxygen saturation (SpO2) of 88–92%, bronchodilators, hydration, systemic corticosteroids, and antibiotics when indicated by signs of bacterial infection, in accordance with current clinical guidelines [25].

**Control group: NIV only.** Patients in the control group will receive only NIV as described above. To ensure consistency and minimize external factors, patients in this group will also wear headphones during the NIV sessions, but no music will be played. The application of headphones in the control group is a deliberate methodological choice aimed at standardizing the experience across both study groups and minimizing potential confounding factors by controlling for the presence of headphones and the procedural aspects.

**Intervention group: NIV combined with music-based intervention.** In the intervention group, patients will receive NIV as detailed in the protocol combined with music-based intervention. The music will be delivered through headphones, allowing patients to experience a soothing auditory environment during the session. Aligning with the 2–4-hour timeframe within which a positive physiological response to NIV is typically observed [4,9], the duration of the music-based intervention is set at two hours. This decision was informed by the results of a preliminary feasibility study conducted on a small cohort of patients. This pilot study demonstrated that a two-hour duration was both well tolerated and effective in maintaining patient comfort and engagement, without undue burden. The optimal duration of music intervention is not yet definitively established in the literature, with previous studies ranging from 10 minutes to 2.5 hours [26]. Furthermore, a previous trial by Messika et al. [16], exploring the effect of a musical intervention on respiratory comfort during NIV in ICU patients with acute respiratory failure, failed to show a statistically significant reduction in respiratory discomfort. The authors suggested that a 30-minute intervention might have been too short, representing a potentially 'under-dosed' music

intervention. Patients will retain control over the intervention, with the option to pause or stop the music at any time by signalling the researcher.

Given the practical limitations of conducting standard psycho-musical assessments for critically ill COPD patients, a qualified music therapist (WJ in the authors' list) created a customized music playlist for this study in consultation with experienced intensivists specializing in respiratory resuscitation (IB and, MB in the authors' list) [27].

This playlist utilizes the 'U' sequence, a standardized receptive music-based intervention technique developed at the University Hospital of Montpellier and designed to induce relaxation through a gradual progression of music. This sequence begins with calming melodies and gradually increases in tempo and intensity, creating a "U-shaped emotional curve" where a downward phase reduces musical elements to promote deep relaxation, followed by an upward re-energizing phase [28,29]. This approach aims to reduce anxiety and promote relaxation in COPD patients experiencing hypercapnic acute respiratory failure undergoing NIV and has also been shown to enhance the caregiver-patient "listening" relationship [30], which is crucial for patient acceptance and adherence to NIV.

To maximize acceptability within a diverse Tunisian cohort (varying in education, musical background, and regional culture), the playlist features familiar and generally well-tolerated Tunisian and Oriental instrumental pieces (oud, qanun, nay, violin), including modal improvisations and a cappella segments suitable for COPD patients generally aged 45 and above. This culturally resonant repertoire incorporates specific musical attributes designed to enhance physiological and emotional well-being, including a slow tempo (60–80 BPM) to align with resting heart rate and promote parasympathetic activation, steady, cyclical rhythms (around 6 breaths/min) to stabilize respiration and improve heart-rate variability, simple consonant maqam improvisations (Nahawand, Ajam, Hajaz, Rast, et Bayati) to minimize cognitive load and facilitate relaxation, and soft instrumental timbres with occasional a cappella to reduce sympathetic activity and cortisol levels. Furthermore, a gradual decrease then increase in dynamics and orchestration (downward then upward U-phase) intends to guide an emotional arc from relaxation to gentle re-activation, potentially supporting NIV acceptance and adherence.

**Strategies to improve adherence to interventions.** To ensure adherence to the intended two-hour music-based intervention, several measures will be implemented. The healthcare professional administering the intervention, part of the core team of senior physicians, will supervise each session. This team will be trained to ensure proper implementation and patient comfort throughout the session. Patients will be positioned semi-recumbently and provided with appropriate headphones. The environment will be kept as quiet as possible, and headphone placement and sound delivery will be regularly checked. Any interruptions due to urgent care or intolerance to NIV will be documented, and the actual duration of exposure to music-based intervention will be recorded for each patient to accurately assess adherence.

## Data collection

**Baseline assessment ($t_0$).** Prior to group allocation, each participant's sociodemographic data (such as age and sex), baseline characteristics including the COPD Assessment Test (CAT), modified Medical Research Council (mMRC) scores, and the Charlson Comorbidity Index (CCI), and prior home NIV use, will be recorded.

Following this assessment, participants will be randomised into one of two study groups.

**During NIV session.** During the first NIV session, primary and secondary outcomes will be assessed by trained residents at 2 and 4 hours using validated instruments, including physiological, neurological and psychological measures.

**Trial implementation and coordination.** The rigorous implementation of this trial relies on strict protocol adherence clearly defined roles and standardized procedures to ensure methodological integrity and internal validity. Prior to enrolment, all ICU medical and paramedical staff will be trained on the study protocol and data collection methods during dedicated briefing sessions.

The randomization sequence will be generated by a research assistant not involved in recruitment, intervention delivery, or outcome assessment. Daily screening and enrolment of eligible participants will be carried out by site investigators. After obtaining written informed consent and confirmation of eligibility, a designated research nurse will perform

the allocation by opening sealed, opaque envelopes containing the group allocation, serving as the sole custodian of the envelopes. Participants assigned to the intervention group will receive a single, two-hour music-based intervention during the first NIV session, administered by a core team of senior physicians. Control group participants will wear headphones without music for the same duration. To ensure protocol fidelity and minimize inter-provider variability, this core team will deliver all intervention sessions according to shift schedules, using a timer to monitor session duration and document key session details. Outcome assessments will be conducted by well-trained residents who are blinded to participants' group assignments and data analysis. To minimize bias, these residents will not be involved in recruitment or intervention delivery. All assessors will undergo specific training to standardize data collection procedures and reduce inter-rater variability.

**Intervention fidelity and data quality assurance.** A standardized protocol governs the delivery of the music-based intervention, ensuring consistency across all sessions. NIV will be managed by ICU staff according to routine clinical care practices. A comprehensive fidelity monitoring plan will be implemented to ensure consistent and reliable delivery of both the intervention and control conditions. This plan addresses key fidelity domains, including adherence, delivery quality, and data accuracy. After each session, senior physicians will complete a fidelity log documenting session duration, deviations, interruptions, patient engagement, and environmental factors.

The music-based intervention will be administered only once per participant, during the first NIV session, ensuring uniform exposure across groups. Sessions lasting less than the intended duration of two hours will be recorded, with reasons documented in the fidelity log. Data quality will be ensured through double data entry and verification processes.

This structured fidelity and quality assurance approach is designed to uphold the trial's internal validity and support accurate interpretation of the intervention's effects.

## Study endpoints

**Primary endpoint.** The primary endpoint is the change in $PaCO_2$ from baseline to 4 hours after the initiation of NIV. $\Delta PaCO_2$, calculated as the difference between $PaCO_2$ value measured at the start of the NIV session ($t_0$) and $PaCO_2$ value recorded at 4 hours ($t_4$): $\Delta PaCO_2 = PaCO_2 (t_0) - PaCO_2 (t_4)$. We hypothesize that the intra-session music-based intervention adjunct to NIV will lead to a significantly greater $\Delta PaCO_2$ compared to standard NIV alone. $PaCO_2$ levels will be measured using the GEM Premier 3500 Blood Gas Analyzer, a device known for its fast, reliable, and efficient blood gas analysis. The GEM Premier 3500 has established validity and reliability for measuring blood gas parameters [31].

**Secondary endpoints.**

- The evolution of $PaCO_2$ during the first 4 hours of NIV. This will be assessed by measuring $PaCO_2$ at baseline, before NIV initiation ($t_0$), at 2 hours ($t_2$), and at 4 hours ($t_4$).

- Unplanned interventions: The number of unplanned interventions required by caregivers during the NIV session will be documented through direct observation and review of nursing notes.

- Changes in respiratory and cardiovascular parameters: Respiratory rate (RR), peripheral oxygen saturation ($SpO_2$), heart rate (HR), systolic blood pressure (SBP), and diastolic blood pressure (DBP) will be continuously monitored and recorded using the Edan Elite V6 modular patient monitoring system. This system provides validated and reliable measurements of these physiological parameters. Tidal volume (Vt) and leak rate will be directly collected from the ventilator's monitoring system (Mindray SV 600 or Löwenstein Elisa 600). Measurements will be taken before NIV initiation ($t_0$), and then at intervals ($t_2$, $t_4$).

- Respiratory comfort: will be assessed using a 100 mm digital visual analog scale (VAS), where 0 represents "no discomfort" and 100 represents "maximum discomfort". Respiratory comfort will be measured at baseline $t_0$, $t_2$ and $t_4$. The VAS is a commonly used and validated tool for subjective symptom assessment [32].

- Neurological assessment ($t_0$, $t_2$, $t_4$) using the following validated scales:
  - Encephalopathy score: The Kelly-Matthay scale, a 5-point scale specifically assessing the level of consciousness in hypercapnic encephalopathy, will be used. Its reliability and validity in this context have been described in the literature [33,34].
  - Richmond Agitation-Sedation Scale (RASS): A validated 10-point scale used to assess the level of sedation or agitation in critically ill patients [35].
- Patient-reported outcomes ($t_0$, $t_2$, $t_4$):
  - Dyspnea will be evaluated using the Borg Dyspnea Scale (BDS):A validated 10-point scale ranging from 0 ("no breathlessness") to 10 ("maximal breathlessness "), reflecting the patient's subjective perception of shortness of breath. The BDS is a well-established and reliable measure of dyspnea [36].
  - Pain intensity will be measured using Verbal Numeric Rating Scale (VNS): A 0-to-10 scale where 0 represents "no pain" and 10 represents "worst possible pain". Patients will verbally rate their current pain level. The VNS is a commonly used and accepted pain assessment tool [37].
- Psychological assessment ($t_0$, $t_4$):
  - Rapid Clinical Test For delirium (4 AT): A brief and validated screening tool for delirium [38].
  - State Trait Anxiety Inventory (STAI): A widely used and validated self-report questionnaire measuring both state (current) and trait (general tendency) anxiety [39].
  - Perceived stress scale (PSS): A widely used and validated 10-item questionnaire measuring the degree to which individuals perceive their lives as stressful [40].

Self-reported questionnaires (STAI and PSS) have been validated for use in Arabic-speak patients [39,40].

Data will be collected using a dedicated case report form. The completed forms will be sent to the data manager for anonymization and entry into a secure database.

The timeline for enrolment, intervention and assessments is outlined in the SPIRIT schedule of events in Table 1.

### Statistical methods

**Sample size.** The sample size calculation was based on the primary outcome, the reduction in arterial $PCO_2$ following NIV. Based on previous studies involving patients with AECOPD treated with NIV, the standard deviation (SD) of $PCO_2$ changes was estimated at 7 mmHg [41]. A clinically significant difference of 5 mmHg in $PaCO_2$ reduction between the group receiving music-based intervention and the control group was considered. To detect this difference with 80% power (1-β) and a two-sided significance level of 5% (α = 0.05), we used the standard formula for comparing two means [42]:

$$n = \frac{2 \times \delta^2 \left( Z_{\frac{\alpha}{2}} + Z_{1-\beta} \right)^2}{\Delta^2}$$

Where:

- $\delta$ represents the standard deviation (SD) of the primary outcome (assumed to be equal across the two groups),

- $\Delta$ the target effect size,

**Table 1. Schedule of enrolment, interventions, and assessments according to SPIRIT 2013 guidelines.**

| | STUDY PERIOD | | | | |
| --- | --- | --- | --- | --- | --- |
| | Enrolment | Allocation | Post-allocation – at NIV session | | Close-out |
| **TIMEPOINT**** | *-t0* | *-t0* | *t0* | *t2* | *t4* |
| **ENROLMENT:** | | | | | |
| **Eligibility screen** | X | | | | |
| **Informed consent** | X | | | | |
| **ALLOCATION** | | X | | | |
| **INTERVENTIONS:** | | | | | |
| *Music-based intervention* | | | ◄━━━━━━━━━━━► | | |
| *Control group* | | | | | |
| **ASSESSMENTS:** | | | | | |
| *Baseline variables:* demographic data, medical history, CCI, obesity status, prior home NIV | X | | | | |
| *Primary outcome:* PCO2 | | | X | | X |
| *Secondary outcomes* | | | | | |
| *Change of PaCO2 over time* | | | X | X | X |
| Respiratory and cardiovascular parameters | | | X | X | X |
| Respiratory comfort | | | X | X | X |
| Neurological assessment (Encephalopathy score, RASS,) | | | X | X | X |
| Psychological assessment (4 AT, STAI, PSS) | | | X | | X |
| Patient reported outcomes (BDS, VNS) | | | X | X | X |
| Unplanned interventions | | | X | X | X |

**CCI:** Charlson Comorbidity Index; **SAPSII:** Simplified Acute Physiology Score II; **BDS:** Borg Dyspnea Scale; **VNS:** Verbal Numeric Rating Scale; **4 AT:** Rapid Clinical Test For delirium; **STAI:** State Trait Anxiety Inventory; **PSS:** Perceived stress scale

- $Z_{\frac{\alpha}{2}}$ and $Z_{1-\beta}$ are the Z-scores corresponding to the chosen significance level (α) and power (1 − β), based on the standard normal distribution.

 Assuming an alpha level of 5% and a power of 80%, a minimum of 31 participants per group is required to detect a statistically significant difference in the reduction of $PCO_2$ between the intervention and control groups. Accounting for a 10% dropout rate [43] the final sample size is set at 35 patients per group. Patients will be randomly assigned to either a 'music-based intervention' group or a 'control' group, with the control group receiving standard NIV care. Recruitment is planned over a 6-month period, with a target enrolment of 70 participants.

 The sample size calculation was performed using Biostat TGV platform, an online tool that relies on the R statistical software for all underlying computations (https://biostatgv.sentiweb.fr/).

 **Statistical analysis.** All statistical analyses will be conducted in the intention-to-treat population at a two-sided 5% alpha risk.

 For each group and at each assessment time, categorical variables will be summarized as numbers and percentages, and continuous variables will be expressed as number, mean, and standard deviation (SD). Quantitative variables with skewed distribution will be presented in terms of median and interquartile range (IQR) (25th-75th percentile). The normality of the distribution will be assessed using the Shapiro-Wilk test. In addition, histograms and Q–Q plots will be examined for each group to visually inspect the distribution.

**Primary endpoint analysis.** The primary outcome is the change in $PaCO_2$, defined for each participant as the difference between baseline ($t_0$) and the value measured immediately after the first NIV session ($t_4$), calculated as: $\Delta PaCO_2 = PaCO_2$ baseline $- PaCO_2$ at 4 hours.

To assess the impact of the music-based intervention, the mean $\Delta PaCO_2$ will be compared between the intervention and control groups using an independent samples t-test, assuming normal distribution and homogeneity of variances. For the primary efficiency analysis, only sessions with the intended duration of 2 hours will be included.

**Adjusted primary analysis.** To account for potential confounding variables, a multivariable linear regression model will be performed, with $\Delta PaCO_2$ as the dependent variable and group allocation (intervention vs. control) as the main independent variable. The model will adjust for baseline factors known or suspected to influence $PaCO_2$ levels, including:

- Dyspnea severity (assessed by the mMRC scale),

- Prior use of home NIV,

- Obesity status,

- Kelly Mathey scale score, and

- Baseline anxiety trait (measured by the STAI trait score).

**Secondary endpoints.**

**Change of $PaCO_2$ over time:** To evaluate the temporal evolution of $PaCO_2$ (at H0, H2, and H4), a linear mixed-effects model (also referred to as multilevel models) will be employed. This statistical approach is particularly suited to repeated measures data, as it accounts for:

- Intra-individual correlation of repeated measurements over time,

- Inter-individual variability (random effects),

- Missing data at some time points, which are common in clinical studies.

This flexible statistical approach is appropriate for examining longitudinal data, handling potentially unbalanced data with missing entries, and accounting for the repeated nature of measurements within each participant [44].

In the model, time (categorical: H0, H2, H4), group (intervention vs. control), and their interaction (group × time) will be included as fixed effects.

A random intercept will be included for each subject to model individual baseline differences.

The group × time interaction term will specifically test whether the change in $PaCO_2$ over time differs significantly between the two groups, providing a complementary view of the intervention's effect beyond the primary endpoint.

- Respiratory and cardiovascular parameters (h0, h2, h4): Differences between groups over time will be analyzed using linear mixed models as for the evolution of PCO2.

- Respiratory comfort (digital visual scale) and patient-reported outcomes (e.g., BDS, VNS) (h0, h2, h4): The evolution of these parameters from baseline (H0) to H4 will be conducted according to the same principles as for the analysis of the evolution of PCO2.

- Neurological and psychological assessments (scales) (h0, h4): Changes from baseline (H0) to the final time point (H4) will be analyzed within groups using paired t-tests. Between-group comparisons of these changes will be performed using independent t-tests.

- Unscheduled interventions: the number of unplanned interventions is a count outcome variable. The comparison groups will therefore be based on a Poisson regression model.

**Sensitivity analysis.** To evaluate whether session duration affects the primary outcome a sensitivity analysis will be performed including all recorded sessions, regardless of length. A separate multivariable linear regression model will be used, with $\Delta PaCO_2$ as the dependent variable, group allocation as the main predictor, and session duration included as an additional covariate. The model will also adjust for the same baseline confounders (mMRC scale, familiarity with home NIV, obesity status, Kelly-Mathey score, and STAI-Trait score). This analysis will help determine whether the duration of the session significantly modifies the effect of the intervention on $PaCO_2$ changes.

**Missing data.** Missing data will be handled using multiple imputation techniques if the proportion of missing data is significant, ensuring unbiased estimates. Results will be interpreted with both statistical significance ($p < 0.05$) and clinical relevance considered.

The statistical analysis will be conducted using SPSS version 20. Analyses will be performed by an independent investigator (DC in the authors' list) blinded to treatment allocation, ensuring unbiased interpretation of the data.

## Ethics approval

The study protocol was prospectively approved by the ethical committee of faculty of medicine of Sousse, Tunisia (CEFMSo_0037_2024) (S2 File).

A Data Monitoring Committee (DMC) will not be established for this trial because the risk to participants is minimal, as music-based intervention is a non-invasive intervention with few or no known adverse effects. NIV is already a standard practice in the management of COPD exacerbations, and the study does not alter its clinical use. Monitoring of adverse events and safety data will be conducted by the research team, in accordance with standard protocols. Any concerns regarding patient safety will be directly reported to the ethics committee.

## Discussion

This trial aims to investigate whether the addition of music-based intervention to standard NIV treatment can improve the efficiency of NIV in patients hospitalized with AECOPDs. We hypothesize that music-based intervention will enhance patient comfort, reduce anxiety, and improve clinical respiratory function. Furthermore, music-based intervention may enhance NIV acceptance by creating a more relaxed and enjoyable treatment experience [18]. By reducing anxiety and promoting a sense of calm, music-based intervention may improve patient adherence to NIV therapy, leading to better treatment outcomes. Moreover, music-based intervention is a complementary, non-invasive, affordable, and enjoyable treatment that is easy to implement.

The findings of this study have the potential to provide valuable insights into the role of music-based intervention in improving the management of AECOPD. If music-based intervention is shown to effectively improve NIV efficiency, it could lead to several positive outcomes. This includes a potential reduction in the duration of hospital stay and the need for more invasive interventions such as intubation and mechanical ventilation. These improvements could translate into significant cost-savings for healthcare systems by optimizing resource utilization and minimizing the burden of prolonged hospitalizations associated with AECOPD exacerbations.

## Supporting information

**S1 File. Standardized Protocol Items: Recommendations for Interventional Trials (SPIRIT) checklist.**
(PDF)

**S2 File. Research Ethics Committee Approval and Project Documents.**
(PDF)

## Acknowledgments

In preparing this work, the authors used ChatGPT 3.5 for language refinement to enhance clarity and coherence. This tool did not generate new content or alter the scientific meaning. The authors reviewed and edited the text and are fully responsible for its content.

## Author contributions

**Conceptualization:** Imen Ben Saida, Wassim Jomaa, Mohamed Boussarsar.

**Methodology:** Imen Ben Saida, Dhekra Chebil, Wassim Jomaa.

**Writing – original draft:** Imen Ben Saida, Marwa Zghidi.

**Writing – review & editing:** Imen Ben Saida, Dhekra Chebil, Khalil Attia, Mohamed Boussarsar.

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
