## [Decision Letter · Decision Letter 0]

Dear Dr. Ben Saida,

Thank you for submitting your manuscript to PLOS ONE. After careful consideration, we feel that it has merit but does not fully meet PLOS ONE’s publication criteria as it currently stands. Therefore, we invite you to submit a revised version of the manuscript that addresses the points raised during the review process.

We look forward to receiving your revised manuscript.

Kind regards,

Tai-Heng Chen, M.D., Ph.D.

Academic Editor

PLOS ONE

Reviewers' comments:

Reviewer's Responses to Questions

**Comments to the Author**

1. Does the manuscript provide a valid rationale for the proposed study, with clearly identified and justified research questions?

Reviewer #1: Partly

Reviewer #2: Partly

Reviewer #3: Partly

2. Is the protocol technically sound and planned in a manner that will lead to a meaningful outcome and allow testing the stated hypotheses?

Reviewer #1: Partly

Reviewer #2: Partly

Reviewer #3: No

3. Is the methodology feasible and described in sufficient detail to allow the work to be replicable?

Reviewer #1: No

Reviewer #2: Yes

Reviewer #3: No

4. Have the authors described where all data underlying the findings will be made available when the study is complete?

Reviewer #1: Yes

Reviewer #2: Yes

Reviewer #3: No

5. Is the manuscript presented in an intelligible fashion and written in standard English?

Reviewer #1: No

Reviewer #2: Yes

Reviewer #3: No

You may also provide optional suggestions and comments to authors that they might find helpful in planning their study.

Reviewer #1: The protocol would benefit greatly from a conceptual framework to guide the music selection, the frequency and length of intervention, the proposed mechanisms of action on pCO2. As written, it is not clear why a 2-hour testing period was selected nor is there sufficient detail about the music selected for intervention (who selected, characteristics, genre, etc). It is not appropriate to use the term music therapy if the intervention was not delivered by a board-certified music therapist who interacted with the participants.

There is no coherent conceptual basis for why music should reduce pCO2 levels among patients with NIV. Justification for the application of headphones only needs to be strengthened.

Efficiency and efficacy of music intervention is used interchangeably; only include one and fully define what is meant by efficiency.

An intervention fidelity monitoring plan would add rigor to this protocol including who is responsible for delivery of the assigned condition, is it one time only or is it repeated, and how data are collected in a reliable and valid manner on all of the outcomes. Details are needed to ensure adherence to a 2-hour intervention period and how shorter sessions are managed with measuring outcomes and how managed in the analysis.

Additional detail are needed to describe the setting from which patients are enrolled (type of unit, number of beds, staffing patterns, etc.).

Details are needed on the how each of the variables of measured and with what specific instruments, including the validity and reliability.

Unclear if the analysis will consider any covariates and how they will be addressed.

Reviewer #2: I suggest some modifications

eligibility : why ph below 7.35 ,I suggest phbelow 7.30 ( in this case niv is mandatory )

kelly scale threshold should be included in the indication for NIV

Why patients in treatment with chronic psycothropic drugs should be excluded?

AECOPD standard therapy should be specified

Ventilator setting : why PEEP should be set at 3-5 cmH2O ( which rationale? )

moreover : which type of ventilators the authors will use ? PS and PEEP should be according to ventilator waves

Reviewer #3: The authors presented the study protocol of a prospective, randomized, single-blinded, parallel-group trial about the effectiveness of music therapy as add on therapy to noninvasive ventilation in acute exacerbation of chronic obstructive pulmonary disease. As primary endpoint variable serves the reduction in arterial Pressure of Carbon dioxide (PCO₂).

The protocol is partly sound from the methodological point of view. The trial is registered but not currently recruiting.

The protocol needs revision from the methodological point of view.

There are methodological issues with the description of the randomization process and the correct coherent statistical analysis taking sample size justification into account. Among others type I error probability, by various sources, is not controlled.

As such the validity and reproducibility of the trial is not given.

detailed comments

L76: The statement can not be directly translated in a hypothesis. Major items of the PICO statement are missing. The classification of the hypothesis, e.g. ∂ equivalence is not given.

L94: The eligibility criteria lack of important exclusions, like age restrictions, pregnancy restriction, dependency restrictions, other trials, etc. This is even not included in the registration.

L112: Process to preserve allocation concealment not clear.

L113: Substitute selection by allocation

L114: "Single randomization list" not clear.

L114_116: Persons responsible and implement are not given. Specification necessary.

L116: Substitute selection by allocation

L124: Statement about implementation of randomization list conflicts to concealment.

L128_131_ Process for implementation not clear.

L131: Reference for PROBE.

L181: Primary endpoint variable unclear. Intervention is not an element of the definition.

If more than one timepoint is used methods to maintain type one error rate need to be specified.

P12: Specify software used for sample size justification.

Give reference for high dropout rate.

Additional sample size increase for change in statistical test conflicts to power, validity and reproducibility of the trial.

P13: Change in statistical analysis procedure based on observed data conflicts to type one error rate. Change from t-test to Mann-Whitney U test implies change in hypotheses. The Shapiro Test has no power. This approach does not maintain the type one error rate., i.e. conflicts to power, validity and reproducibility of the trial.

State the planned statistical model with parameters for primary endpoint variable analysis!

P14: Substitute bilateral by two-sided

P14: methods against bias are not referred to.

**Do you want your identity to be public for this peer review?** For information about this choice, including consent withdrawal, please see our Privacy Policy

Reviewer #1: No

Reviewer #2: **Yes: ** Antonello Nicolini

Reviewer #3: No

---

## [Author Response · Author response to Decision Letter 1]

9 May 2025

We would like to express our sincere gratitude to the editorial board and the reviewers for their time and careful consideration of our manuscript entitled "Efficiency of a music-based intervention as an adjunct to the first noninvasive ventilation session in acute exacerbation of COPD: a randomized single-blind controlled trial protocol".

We have made all necessary changes to the manuscript, taking into account the reviewers' comments.

We have also attached a rebuttal letter ("Response to Reviewers") that addresses each point raised.

---

## [Decision Letter · Decision Letter 1]

Efficiency of a music-based intervention as an adjunct to the first noninvasive ventilation session in acute exacerbation of COPD: a randomized single-blind controlled trial protocol

PONE-D-25-06120R1

Dear Dr. Ben Saida,

We’re pleased to inform you that your manuscript has been judged scientifically suitable for publication and will be formally accepted for publication once it meets all outstanding technical requirements.

Kind regards,

Tai-Heng Chen, M.D., Ph.D.

Academic Editor

PLOS ONE

Additional Editor Comments (optional):

Reviewers' comments:

Reviewer's Responses to Questions

**Comments to the Author**

1. Does the manuscript provide a valid rationale for the proposed study, with clearly identified and justified research questions?

Reviewer #3: Yes

2. Is the protocol technically sound and planned in a manner that will lead to a meaningful outcome and allow testing the stated hypotheses?

Reviewer #3: Yes

3. Is the methodology feasible and described in sufficient detail to allow the work to be replicable?

Reviewer #3: Yes

4. Have the authors described where all data underlying the findings will be made available when the study is complete?

Reviewer #3: Yes

5. Is the manuscript presented in an intelligible fashion and written in standard English?

Reviewer #3: Yes

You may also provide optional suggestions and comments to authors that they might find helpful in planning their study.

Reviewer #3: I do not have any further comments. All my concerns are considered. The paper is read for publication.

**Do you want your identity to be public for this peer review?** For information about this choice, including consent withdrawal, please see our Privacy Policy

Reviewer #3: No

---

## [Editor Report · Acceptance letter]

PONE-D-25-06120R1

PLOS ONE

Dear Dr. Ben Saida,

I'm pleased to inform you that your manuscript has been deemed suitable for publication in PLOS ONE. Congratulations! Your manuscript is now being handed over to our production team.

Kind regards,

on behalf of

Dr. Tai-Heng Chen

Academic Editor

PLOS ONE